# Solid–Liquid Europium Ion Extraction via Phosphonic Acid-Functionalized Polyvinylidene Fluoride Siloxanes

**DOI:** 10.3390/polym12091955

**Published:** 2020-08-28

**Authors:** Mohammad Wehbi, Ahmad Mehdi, Ali Alaaeddine, Nada Jaber, Bruno Ameduri

**Affiliations:** 1ICGM, Univ. Montpellier, CNRS, ENSCM, 34095 Montpellier, France; ahmad.mehdi@umontpellier.fr; 2Department of Chemistry and Biochemistry, Faculty of Sciences 1, Lebanese University, Rafic Hariri Campus—Hadath, Beirut 6573/14, Lebanon; Alikassem.alaaeddine@ul.edu.lb (A.A.); nada.jaber@ul.edu.lb (N.J.)

**Keywords:** radical copolymerization, fluoropolymers, europium extraction, sol–gel, phosphonates

## Abstract

Novel triethoxysilane and dimethyl phosphonate functional vinylidene fluoride (VDF)-containing terpolymers, for potential applications in Eu ion extraction from water, were produced by conventional radical terpolymerization of VDF with vinyltriethoxylsilane (VTEOS) and vinyldimethylphosphonate (VDMP). Although initial attempts for the copolymerization of VTEOS and VDMP failed, the successful terpolymerization was initiated by peroxide to lead to multiple poly(VDF-*ter*-VDMP-*ter*-VTEOS) terpolymers, that had different molar percentages of VDF (70–90 mol.%), VTEOS (5–20 mol.%) and VDMP (10 mol.%) in 50–80% yields. The obtained terpolymers were characterized by ^1^H, ^19^F, ^29^Si and ^31^P NMR spectroscopies. The crosslinking of such resulting poly(VDF-*ter*-VDMP-*ter*-VTEOS) terpolymers was achieved by hydrolysis and condensation (sol–gel process) of the triethoxysilane groups in acidic media, to obtain a 3D network, which was analyzed by solid state ^29^Si and ^31^P NMR spectroscopies, TGA and DSC. The thermal stability of the terpolymers was moderately high (up to 300 °C under air), whereas they display a slight increase in their crystallinity-rate from 9.7% to 12.1% after crosslinking. Finally, the dimethyl phosphonate functions were hydrolyzed into phosphonic acid successfully, and the europium ion extraction capacity of terpolymer was studied. The results demonstrated a very high removal capacity of Eu(III) ions from water, up to a total removal at low concentrations.

## 1. Introduction

The potential risks of pollution by metal ions in aqueous environmental is still a major environmental issue, due to their high toxicity, persistence and potential carcinogenic effects [1]. Thus, the removal of such ions is vital to the environment and has led to various studies [2] focusing on the development of methods and techniques for their extraction from heterogeneous media such as liquid–liquid extraction and solid-phase extraction [3,4,5,6], chemical precipitation [7], membrane filtration [8], flotation [9], electrochemical [10] and biofiltration methods [11]. 

Solid-phase extraction (SPE), in particular, where ion exchange or chelating materials are employed for metal ion separation, has attracted great interest, due to its simplicity, cost effectiveness, and reduced solvent use [12]. The concept of polymers bearing chelating functions for the employment as SPE has been reported [13]. Several approaches have been studied to obtain these functional polymers such as impregnation [14], grafting of the desired functional groups [15,16], and copolymerization of functional monomers with curing site monomers [17,18,19,20]. 

Recently, phosphorus-containing monomers and polymers [21,22,23,24,25] have triggered a great interest, since they can be involved in many applications, including as corrosion inhibiting agents [26,27], flame retardants [28], adhesion promoters for paints, superlubricity coatings [29], water repellent [30], polymer electrolyte membrane fuel cells [31,32,33,34], and in biomedical fields [25]. In addition, due to their excellent chelating properties [35], abundant examples of phosphorus functional polymers are present for employment as SPEs. Kabanov et al. [36] prepared crosslinkable polymers from the copolymerization of diethyl vinyl phosphonate and acrylic acid with *N,N*-methylene diacrylamide for extracting Cu^2+^ ions. Goto et al. [37] synthesized polymers based on the polymerization of 1,12-dodecanediol-O,O’-diphenyl-phosphonic acid (DDDPA) with divinylbenzene to successfully extract zinc ions from aqueous media. Inspired by Goto’s work, Zhu et al. [38] reported the emulsion polymerization of DDDPA and 4-vinylpyridine to prepare polymers with a high selectivity to lead ions.

Fluorinated polymers are specialty polymers involved in many applications such as aeronautics, aerospace and petrochemical industries, as well as textiles and fabrics, optics, chemical engineering, microelectronics, automotive, and building industries [39,40,41]. To fully take advantage of their unique properties, recent studies have focused on their development and incorporation in different kinds of consumables and products. One of the major members of this polymer family is poly(vinylidene fluoride), PVDF, which is the second most produced fluoropolymer after polytetrafluoroethylene (PTFE) [41,42,43,44]. This is because PVDF almost displays the same properties as those of PTFE, while its starting monomer, vinylidene fluoride (VDF), is much less dangerous than TFE (not explosive and with a lower toxicity) and also shows a comparable reactivity [45]. Moreover, PVDF has found applications in coatings, piezoelectrical devices (as actuators), binders for Lithium ion batteries, and treatment of wastewater. Zhao et al. [46,47] reported the preparation of chelating membranes based on melamine-diethylenetriaminepentaacetic acid/PVDF (MA–DTPA/PVDF) and bearing polyaminecarboxylate groups for Ni(II) removal from wastewater. In addition, Wang et al. [48] described the Pd/Fe chelation from polyacrylic acid/PVDF membranes. Both teams indicated that the hydrophobicity of the copolymers (triggered from the fluorine atoms) led to an easier separation and material retrieval from the solution. Various investigations deal with the functionalization of PVDF-based membranes by multiple functional groups for wastewater treatment [49,50,51,52], but only one recent study has reported the prominence of phosphorus functional PVDF involved in Rare Earth metal extraction in strong acidic media [53,54]. 

To achieve a high durability and material reusability, it is required that polymers involved as SPEs be crosslinked. Furthermore, this crosslinking induces the heterogeneity of these materials in complex aqueous media. The traditional methods for crosslinking VDF-based copolymers range from the usage of (i) diamines, (ii) bisphenols, (iii) via peroxides/trienes, to (iv) radioactivation [55]. However, it would be of interest to find a more simple technique to crosslink PVDF. Actually, triethoxysilane groups can offer a simpler and more elegant strategy for such crosslinking via the sol–gel process [56]. Several studies have been reported regarding triethoxysilyl-functionalized fluoropolymers, mainly for improved adhesion [57], oil and water repellencies [58,59,60], solvent resistance [61], fuel cell membranes [62] and for amine functionalization [63,64,65,66]. Earlier, Asandei et al. [67] attempted some copolymerizations of VDF with vinyltriethoxylsilane in low yields.

Although multiple investigations have described the use of phosphonate-containing mesoporous materials for the extraction of rare earth metals [68], lanthanides and actenides [69,70,71,72], it would be of interest to test potential synergistic effects of phosphorus, fluorine and silicon atoms for heavy metal extraction. Thus, the objective of this manuscript aims at highlighting the preparation of novel PVDF-bearing phosphonate and triethoxysilyl, poly(VDF-*ter*-VTEOS-*ter*-VDMP) terpolymers, from the radical terpolymerization of commercially available monomers such as VDF, dimethylvinylphosphonate and vinyltriethoxysilane using a peroxide initiator. Detailed NMR (^1^H, ^19^F, ^31^P and ^29^Si) spectroscopy of the purified terpolymers was obtained before and after crosslinking as well as their thermal properties. The resulting materials were used to extract europium ions from aqueous media. The choice of europium as a representative of lanthanides is due to its possessing similar behavior as other lanthanides during extraction [73,74] and because it easily forms complexes with xylenol orange [75,76], a commonly used indicator in colorimetric titrations [77]. 

## 2. Experimental Section 

### 2.1. Materials 

All reagents were used as received unless stated otherwise. Vinylidene fluoride (1,1-difluoroethylene, VDF) was kindly offered by Arkema (Pierre Benite, France). *Tert*-butyl peroxypivalate (TBPPi, purity 75%), *tert*-amyl peroxy-2-ethylhexanoate (TAPE, purity 95%), and 2,5-dimethyl-2,5-di(*tert*-butylperoxy) hexane (DTBPH, purity 90%) were purchased from AkzoNobel (Chalons sur Marne, France). Europium (III) chloride hexahydrate (Purity 99.9%) and vinyltriethoxylsilane (VTEOS, purity 97%) were bought from Aldrich (Aldrich Chimie, 38299 Saint Quentin-Fallavier, France). Vinyldimethylphosphonate (VDMP, purity 90%) was supplied by ABCR (Karlsruhe, Germany). Reagent Plus grade pentane, 3,3′-bis[N,N-bis(carboxymethyl)aminomethyl]-o-cresolsulfonephthalein tetrasodium salt (Xylenol orange), dimethyl carbonate (DMC, purity > 99%), (ethylenedinitrilo)tetraacetic acid tetrasodium Salt (EDTA), and hydrochloric acid were purchased from Sigma-Aldrich and used as received. Deuterated dimethylsulfoxide (DMSO-*d*_6_), acetone and dimethylformamide (DMF-*d*_7_) (purity > 99.8%), used for recording NMR spectra, were supplied from Euroiso-top (Grenoble, France).

### 2.2. Characterization 

#### 2.2.1. Nuclear Magnetic Resonance (NMR) Spectroscopy 

The compositions and microstructures of the copolymers were determined by ^19^F, ^1^H, ^29^Si and ^31^P NMR spectroscopies, recorded on a Bruker AC 400 Spectrometer (376 MHz for ^19^F, 400 MHz for ^1^H, 162 MHz for ^31^P, and 80 MHz for ^29^Si) using DMSO-*d*_6_ as a solvent. Chemical shifts and coupling constants are given in parts per million (ppm) and Hertz (Hz), respectively. 

#### 2.2.2. Thermogravimetric Analysis (TGA) 

The thermogravimetric analysis of the purified co- and terpolymer was performed under air using a TA Instruments TGA 51 apparatus at a heating rate of 10 °C min^−1^ from 24 °C to 580 °C.

#### 2.2.3. Differential Scanning Calorimetry (DSC)

DSC thermograms of the poly(VDF-*ter*-VDMP-*ter*-VTEOS) terpolymers were recorded from a Netzsch DSC 200 F3 instrument under nitrogen. The DSC apparatus was calibrated with noble metals (e.g., indium sample (*T_m_* = 156 °C) before analysis. The heating or cooling ranges were from −40 °C to 200 °C at a scanning rate of 10 °C min^−1^. Melting points were assessed from the maximum of the enthalpy peaks and its area led to the melting enthalpy (Δ*H_m_*).

The crystallinity rates of the co- and terpolymers were calculated from Equation (1):(1)$$Crystallinity-rate\;\left(\chi \right)=\;\;\frac{\Delta H_{m}}{\Delta H_{c}}\;\times 100$$
where Δ*H_c_* (104.5 J g^−1^) and Δ*H_m_* stand for the enthalpy of melting of a 100% crystalline PVDF [78,79] and the heat of fusion (assessed by DSC in J g^−1^), respectively. 

### 2.3. Radical Terpolymerization of VDF with VTEOS and VDMP

The radical terpolymerization of VDF with VDMP and VTEOS (**P_20_**, Table 1) was carried out in a 100 mL HC 276 Hastelloy autoclave Parr system HC 27 equipped with a mechanical Hastelloy anchor, a manometer, a rupture disk (3000 PSI), inlet and outlet valves, and a Parr electronic controller (for regulating the stirring speed and heating). Before the reaction started, the absence of any leaks was checked by pressurizing the autoclave with 30 bar nitrogen. Then, it was placed under vacuum (10^−4^ bar) for 30 min to remove any residual oxygen traces. A solution composed of DTBPH (0.77 g, 2.7 mmol), VTEOS (10.20 g, 53.5 mmol), and VDMP (3.60 g, 26.7 mmol) in 70 mL of DMC was degassed by nitrogen bubbling for 30 min. This solution was introduced into the autoclave under vacuum by means of a funnel connected tightly to the introduction valve of the autoclave. Afterwards, the reactor was cooled in a liquid nitrogen bath, and VDF gas (12.00 g, 187.5 mmol) was transferred into it under weight control. Then, the autoclave was mechanically stirred and gradually heated up to 115 °C, while the evolutions of temperature and pressure (P_max_ = 36 bar (**P_20_**, Table 1)) were recorded. After 16 h, the reaction was stopped (P_min_ = 11 bar) by placing the autoclave in an ice bath. The unreacted gaseous monomer was vented off before opening the autoclave and its content was quickly transferred into a Schlenk flask, where the solvent and unreacted liquid monomer (if there was any) were completely removed under vacuum, avoiding the exposure of the polymer to moisture. Then, the total product mixture was dissolved in a minimum amount of dry acetone and chilled pentane was added through a cannula under vacuum to precipitate the polymer, and then dried under vacuum (20 × 10^−3^ bar, 50 °C) for 24 h. The poly(VDF-*ter*-VDMP-*ter*-VTEOS) terpolymer, as a white wax, was obtained in 80% yield and characterized by ^1^H, ^19^F, ^31^P and ^28^Si NMR spectroscopies. 

^1^H NMR (400 MHz, DMF-*d*_7_, *δ* ppm of **P_20_**, Table 1, Figure 1): 0.40 to 0.70 (m, 1H, −CF_2_CH_2_−CH_2_CH−Si(OCH_2_CH_3_)_3_), 1.12 to 1.31 (m, 9H, −CF_2_CH_2_−CH_2_CH−Si(OCH_2_CH_3_)_3_), 1.70 (m, 1H, −CF_2_CH_2_−CH_2_CH−PO(OCH_3_)_2_), 1.90 (−CH_2_CH−(Si(OCH_2_CH_3_)_3_) and (−CH_2_CH−(PO(OCH_3_)_2_)), 2.15 to 2.40 (m, −CF_2_CH_2_−CH_2_CF_2_− reverse VDF−VDF T-T dyad addition); 2.70 to 3.10 (m, −CH_2_CF_2_−CH_2_CF_2_−, normal VDF−VDF H-T dyad addition), 3.40 (s, 3H, PVDF-CH_2_OC(O)OCH_3_ from DMC radical that initiates a PVDF chain), 3.65 (m, 6H, −CF_2_CH_2_−CH_2_CH−Si(OCH_2_CH_3_)_3_); 3.70 (s, 6H, −CF_2_CH_2_−CH_2_CH−PO(OCH_3_)_2_); 4.40 (m, 2H, PVDF-CH_2_OC(O)OCH_3_ from DMC radical that initiates a PVDF chain), 6.05 to 6.45 (tt, ^2^*J*_HF_ = 55.0 Hz, ^3^*J*_HH_ = 4.6 Hz), −CH_2_CF_2_−H end-group originated from the transfer of proton to solvent or polymer or from the back biting [80].

^19^F NMR (376 MHz, DMF-*d*_7_, *δ* ppm of **P_20_**, Table 1, Figure 2): −91.50 to −93.50 (−CH_2_CF_2_−CH_2_CF_2_−normal Head-to-Tail VDF−VDF dyad); −94.50 (–CF_2_ of VDF in the VDF–VDMP/VTEOS dyad); −114.00 (−CH_2_CF_2_−CF_2_CH_2_−CH_2_, reverse Head-to-Head VDF−VDF dyad); −115.0 (small dtt, ^2^*J*_HF_ = 55.00 Hz, ^3^*J*_HF_ = 16.00 Hz and ^4^*J*_FF_ = 6 Hz, CF_2_-CH_2_CF_2_-H, chain-end from transfer); −116.50 (−CH_2_CF_2_−CF_2_CH_2_−CH_2_, reverse H-H VDF−VDF dyad).

^31^P NMR (162 MHz, DMF-*d*_7_, *δ* ppm of **P_20_**, Appendix A): singlet centered at 37.50 ppm assigned to −P(O)(OCH_3_)_2_.

^29^Si NMR (80 MHz, DMF-*d*_7_, *δ* ppm of **P_20_**, Appendix A): −44.00 (s, -Si(OCH_2_CH_3_)_3_ adjacent to normal H-T VDF-VDF dyad), −46.00 (s, -Si(OCH_2_CH_3_)_3_) adjacent to reverse H-H VDF−VDF dyad).

### 2.4. Crosslinking of poly(VDF-ter-VDMP-ter-VTEOS) Terpolymer 

The crosslinking of poly(VDF-*ter*-VDMP-*ter*-VTEOS) terpolymer was performed by hydrolysis and polycondensation of the dangling triethoxysilyl groups on the polymers backbone. Typically, on a vortex mixer, 5.00 g of terpolymer were dissolved in DMF. Then, 1 drop of hydrochloric acid (12 M) was added and the mixture was stirred immediately for a couple of seconds and then kept stationary to avoid any disturbance for the network formation during the sol–gel process. After 2 h, a milky gel was produced and kept to age for one week at 24 °C, followed by three washings with acetone prior to solvent removal under reduced pressure (20 × 10^−3^ bar, 80 °C) for 24 h. The final obtained product, a yellowish resin, was characterized by solid state ^31^P and ^29^Si NMR spectroscopies. 

One Pulse–Magic angle spinning (OP-MAS) Solid state ^31^P NMR (162 MHz, *δ* ppm of **P_20_** after crosslinking, Figure 6): −24 (–PO(OH)_2_), −35 (–PO(OMe)(OH)) and −44 (–PO(OMe)_2_).

OP-MAS Solid state ^29^Si NMR (80 MHz, *δ* ppm of **P_20_** after crosslinking, Appendix A): −57 (T^2^–Si(OH)(OSi)_2_ and −67 (T^3^–Si(OSi)_3_). 

### 2.5. Hydrolysis of Phosphonate Groups into Phosphonic Acid in poly(VDF-ter-VDMP-ter-VTEOS) Terpolymer 

The hydrolysis of dimethyl phosphonate groups in poly(VDF-*ter*-VDMP-*ter*-VTEOS) terpolymer was achieved following the method reported by McKenna et al. [81] Under N_2_ purging at 24 °C, a round-bottom flask containing poly(VDF-*ter*-VDMP-*ter*-VTEOS) terpolymer (2.00 g) was mixed in DMF (20 mL) until it swells. The polymer solution was then mixed with 3 bromotrimethylsilane equivalents, dropwise added under stirring for 1 h. The reaction proceeded for further 16 h at 24 °C, followed by its quenching with 10 MeOH equivalents. After filtration, the polymer was dried under vacuum (20 × 10^−3^ bar, 50 °C) for 16 h to yield poly(VDF-*ter*-VPA-*ter*-VTEOS) terpolymer (2.0 g of brown resin). The hydrolyzed product was characterized by solid state ^29^Si (Appendix A) and ^31^P (Figure 9) NMR spectroscopies.

### 2.6. Complexometric Titration

Ion exchange capabilities of poly(VDF-*ter*-VPA-*ter*-VTEOS) terpolymers were assessed by complexometric titration. The ion exchange process was achieved by the introduction of poly(VDF-*ter*-VPA-*ter*-VTEOS) terpolymer into aqueous solutions of Eu(III) at several initial concentrations (2, 4, 6, 8 and 10 × 10^−3^ mol·L^−1^). Lanthanide solutions were prepared by dissolving the appropriate europium (III) chloride hexahydrate amount into a mixture composed of 15 mL acetic buffer and 10 mL water (pH 5.8). 0.250 g of the terpolymers was then poured into each lanthanide solution and stirred for 24 h to enable the ion exchange. The filtrates were then titrated, using a buret, with EDTA (0.01 M), in which a few drops of xylenol orange (indicator) were added. The EDTA addition ceased when the color of the solution turned from violet to yellow. The uptake (α%) was calculated by Equation (2): (2)$$\alpha \%=\frac{Ci-Ce}{Ci}\times \;100$$
where Ci and Ce stand for the initial and remaining Eu(III) concentrations in the solution before and after the ion exchange, respectively.

## 3. Results and Discussion

### 3.1. Preparation of poly(VDF-ter-VDMP-ter-VTEOS) Terpolymers

Poly(VDF-*ter*-VDMP-*ter*-VTEOS) terpolymers were synthesized by conventional radical terpolymerization of vinylidene fluoride (VDF) with vinyldimethylphosphonate (VDMP) and vinyltriethoxylsilane (VTEOS) at 115 °C in dimethyl carbonate (DMC) as a solvent and initiated by 2,5-dimethyl-2,5-di(*tert*-butylperoxy) hexane (DTBPH) (Scheme 1). 

The polymerization took place in a high-pressure autoclave, since VDF is a gas. Various reaction conditions were explored (Table 1) such as different reaction conditions: (i) solvents (dimethyl carbonate or 1,1,1,3,3-pentafluorobutane), (ii) temperatures (57 to 135 °C depending on the used initiator), (iii) initiators (*tert*-butyl peroxypivalate (TBPPi at 57 °C), *tert*-amyl peroxy-2-ethylhexanoate (TAPE at 73 °C), and 2,5-dimethyl-2,5-di(*tert*-butylperoxy) hexane (DTBPH between 115 and 135 °C)), and (iv) varying the initial monomer ratios in the reaction feeds. It was first decided that the copolymerizations of different monomers were attempted. The copolymerization of VTEOS and VDMP (**P_1–3_**) failed whatever the used initiator or reaction temperature. In addition, the radical copolymerization of VDF with either VTEOS or VDMP also failed when TAPE or TBPPi initiated the reactions (**P_4–6_**, **P_11–12_**, **P_16–17_**), while the copolymerization was successful when DTBPH was chosen as the initiator (**P_7–10_**, **P_13–15_**, **P_18–20_**). 

Indeed, similar to TAPE and TBPPi, DTBPH is a peroxide, but the copolymerization of VDF and VTEOS and the terpolymerization occurred only when that last one was used. Considering the DTBPH structure and compared to monofunctional TBPPi and TAPE, this initiator is difunctional, meaning that the radical density it releases into the reaction medium is double to that of the other initiators for a similar molar percentage [82] (Appendix A). This increase in radical density ensures that this copolymerization could be initiated more efficiently.

Moreover, the higher dissociation temperature of its O-O bond (the half-life is 10 h at 115 °C compared to 57 °C and 75 °C for TBPPi and TAPE, respectively) leads to a higher overall pressure in the reactor, hence producing a higher reactivity of VDF and thus a more successful terpolymerization. 

Similar results were noted when the copolymerization of VDF with VDMP was performed (**P_13–15_**), which confirms that DTBPH is a suitable initiator. It is worth observing that using another solvent of polymerization did not affect the reaction. Thus, we decided to try dimethyl carbonate (DMC) since it is considered to be a green solvent [83,84,85] and swells PVDF well [83]. However, changing the reaction temperature between 115 and 135 °C enhanced the yields of the overall reaction (75% at 115 °C (**P_10_**) compared to 62% at 135 °C (**P_7_**)) which may be explained by the half-lives of the initiator (10 h at 115 °C while it is 1 h at 135 °C) and its ability to release radicals for an extended amount of time at lower temperature. Based on the obtained results (vide infra), poly(VDF-*ter*-VDMP-*ter*-VTEOS) terpolymers were successful synthesized. Actually, the presence of VDF in the reaction medium allows the terpolymerization of VTEOS and VDMP, a monomer pair that failed in copolymerization. Thus, this polymerization reaction is considered to be a “termonomer-induced copolymerization” [43].

### 3.2. Characterization of poly(VDF-ter-VTEOS-ter-VDMP) Terpolymers by ^1^H, ^19^F, ^31^P and ^29^Si NMR Spectroscopies 

Purified poly(VDF-*ter*-VTEOS-*ter*-VDMP) terpolymers were analyzed by ^1^H, ^19^F, ^31^P, and ^29^Si NMR spectroscopies. The ^1^H NMR spectrum (Figure 1) of the poly(VDF-*ter*-VDMP-*ter*-VTEOS) terpolymer (**P_20_**) exhibits eight characteristic signals, mainly at: (i) 0.40 to 0.70 ppm assigned to the methine proton adjacent to the Si atom, (ii) 1.10 to 1.30 ppm for the three CH_3_ groups of the triethoxysilyl units, (iii) 1.70 ppm corresponding to the CH group adjacent to the PO group, (iv) 1.90 ppm singlet of the CH_3_ protons in the dimethyl phosphonate side group, (v) 2.15 to 2.40 ppm range attributed to the reverse (tail-to-tail, T-T) addition of VDF repeat units (−CF_2_CH_2_−CH_2_CF_2_−) [83,86,87,88,89,90,91,92], (vi) 2.70 to 3.10 ppm assigned to normal (head-to-tail, H-T) addition of VDF (−CH_2_CF_2_−CH_2_CF_2_−) [83,86,87,88,89,90,91,92], (vii) 2.80 ppm for protons of ethylene in VDF and VTEOS or VDMP dyad, and finally (viii) 3.65 ppm corresponding to the CH_2_ of triethoxysilane group. However, this last signal overlaps with that of DMC which cannot be removed due to the highly viscous terpolymers. The small triplet (^2^J_HF_ = 45 Hz) of triplets (^3^J_HH_ = 7.0 Hz), centered at 6.30 ppm, corresponds to the end-proton in –CH_2_CF_2_–H, and suggests back-biting [80] or transfers to monomers, solvent or copolymer. 

The ^19^F NMR spectrum displays the expected signals of PVDF (**P_20_**, Figure 2), centered between −91.50 to −93.50 ppm attributed to the normal VDF−VDF H-T dyad while that at −94.50 ppm is assigned to VDF in the alternating VDF–VDMP/VTEOS dyad, a peak of small intensity at −107.50 ppm corresponding to CH_3_CF_2_CF_2_CH_2_– [93], and signals between −114.00 and −116.50 ppm for reverse VDF−VDF H-H dyad. 

The ^31^P NMR spectrum (Appendix A) of **P_20_** (Table 1) features only a single signal at 37.50 ppm characteristic of the dangling phosphonate group in the terpolymer.

The ^29^Si NMR spectrum (**P_20_**
Table 1, Appendix A) shows two signals: that centered at −44.00 ppm is assigned to the triethoxysilane group next to a head to tail VDF dyad, while the second one at −46 ppm corresponds to a triethoxysilane group (−Si(OCH_2_CH_3_)_3_) adjacent to a reverse head to head VDF-VDF dyad. The ^29^Si NMR spectrum does not display any signal in the −50.00 and −70.00 ppm range indicating that no crosslinking of the dangling silyl groups took place yet. 

### 3.3. Crosslinking of poly(VDF-ter-VDMP-ter-VTEOS) Terpolymer

The crosslinking of poly(VDF-*ter*-VDMP-*ter*-VTEOS) terpolymers was performed via the hydrolysis and condensation of the pendant triethoxysilyl groups. The sol–gel transformation makes it possible to prepare a 3D network by the formation of strong Si–O–Si bonds. Usually, such a crosslinking requires H^+^, OH^−^ or a nucleophile such as a fluoride anion or a Lewis base which acts as a catalyst [94]. First, as it is an efficient source of nucleophilic F^−^ [95], tetrabutylammonium fluoride (TBAF)/water system was used to crosslink poly(VDF-*ter*-VDMP-*ter*-VTEOS) terpolymers in DMF. However, its addition induced some color turning black leading to an unsuccessful gelation. This stems from a dehydrofluorination of highly acidic protons in PVDF backbone (i.e., CF_2_C*H*_2_CF_2_) in the presence of such a nucleophilic F^−^ instead of yielding to the hydrolysis of triethoxysilyl groups [96]. To overcome this issue, as PVDF is stable in acids [45,85], highly concentrated HCl (12 M) was used as a catalyst. This strong acid was chosen because of the effect of pH on the fast hydrolysis and condensation processes of ethoxysilanes, as depicted in Figure 3.

Actually, at acidic pH, the hydrolysis step occurs rapidly, while the condensation is slow. However, strong acidic conditions (pH < 2) enable a fast condensation that favors the polymer to crosslink [97].

This was observed by a rapid gel formation upon adding one HCl drop into the terpolymer DMF solution (Figure 4). The gel was kept for one week to favor a complete crosslinking. All tested terpolymers containing 5, 10, and 20% triethoxysilane were able to form stable gels, but it was noted that the strength of the gel depends on the triethoxysilane percentage, for which the higher their amount, the stronger the gel formed. After reaction, the solvent was removed at 80 °C under vacuum to yield a solid orange-yellow powder (Figure 4). 

The obtained resins were characterized by ^29^Si and ^31^P solid state NMR spectroscopies. The ^29^Si NMR (Appendix A) spectrum displays two broad signals: the first one centered at −57.00 ppm assigned to T^2^ Si(OH)(OSi)_2_ substructure while that at −67.00 ppm corresponds to T^3^ substructure -Si(OSi)_3_. One would expect to obtain T^3^ substructure only, but, as the crosslinking amount increases in the condensation reaction, the polymer chains lose their flexibility and hence, are unable to fold to produce the total condensation of triethoxysilyl groups (Figure 5). 

The ^31^P NMR spectrum (Figure 6) exhibits three signals: (i) that at −24.00 ppm is attributed to a complete hydrolysis of phosphonate groups into phosphonic acid functions (-PO(OH)_2_), (ii) the one at −35.00 ppm corresponds to partially hydrolyzed phosphonate groups (i.e., -PO(OMe)(OH)), and (iii) that at −44.00 ppm to unreactive dimethyl phosphonate groups [31]. Such a hydrolysis of phosphonate groups into phosphonic acid and partially hydrolyzed phosphonate results from the strong acidic conditions during the crosslinking. However, this is not an issue, since the desired final product is a functional terpolymer bearing phosphonic acid, namely poly(VDF-*ter*-VPA-*ter*-VTEOS) terpolymer.

### 3.4. Thermal Properties of poly(VDF-ter-VTEOS-ter-VDMP) Terpolymer

The thermal properties of such terpolymers were studied by thermogravimetric analysis (TGA) and differential scanning calorimetry (DSC). First, the thermal stabilities of poly(VDF-*ter*-VDMP-*ter*-VTEOS) terpolymers before and after crosslinking were determined by TGA under air (Figure 7). Before crosslinking, an initial loss was noted at 130 °C attributed to the loss of the trapped solvent (DMC) in the polymer chains. A second degradation also occurred from 300 °C mainly assigned to the decomposition of the fluorinated polymer backbone [99,100,101], which continues up to the total terpolymer degradation at about 580 °C. After crosslinking, a single degradation was observed at 180 °C, which is much steeper than that of the uncrosslinked terpolymer, up to 400 °C. This is surprising since the crosslinking is expected to enhance the thermal properties of the terpolymer, which is only noted above 400 °C, where no more degradation happened, while 20% of the residue still remains. This residue may be the thermally stable siloxane units arising from the crosslinking of triethoxysilyl groups. 

The melting temperatures (T_m_) of the terpolymers before and after crosslinking were determined by DSC (Appendix A) and their crystallinity rates were assessed from Equation (1) [78,79].

As expected for PVDF or copolymers containing a high amount of VDF, the glass transition temperature, Tg, was not detected by DSC. However, no significant difference in the T_m_ values before (133 °C) and after (136 °C) crosslinking of the terpolymer was noted. As for the crystallinity rate (χ), a slight increase was observed from 9.7% to 12.1% after crosslinking, maybe related to two parameters: (i) the formation of a stiff non-flexible network that may slightly increase the terpolymer crystallinity or (ii) several oligo(VDF) hydrophobic zones that gather with each other.

### 3.5. Hydrolysis of Phosphonate Groups into Phosphonic Acid in poly(VDF-ter-VTEOS-ter-VDMP) Terpolymer 

During the crosslinking terpolymers, the phosphonate groups were hydrolyzed into phosphonic acids using bromotrimethylsilane [54] at 24 °C (Figure 8) (the detailed procedure is supplied in the Experimental Section), using an improved procedure reported by McKenna et al. [81]. The solid state ^31^P NMR spectrum (Figure 9) indicates that the total hydrolysis of the phosphonate groups was not possible, as evidenced by the presence of two signals centered at 44 ppm (corresponding to the remaining dimethyl phosphonate group) and at 25 ppm (attributed to the totally hydrolyzed phosphonic acid groups). Similar results were obtained when highly concentrated HCl (12 M) was used while heating at 60 °C was attempted. The signal assigned to phosphonic acid appears broader, with a shoulder on one side that is probably due to the hydrogen bonding of P-OH groups. The deconvolution of the signals in the NMR spectrum enabled us to assess the amount of each phosphorus species showing that 69% of the phosphonate species was fully hydrolyzed. This may arise from the inaccessibility of phosphonate groups embedded inside the terpolymer network. 

After reaction with bromotrimethylsilane, the ^29^Si NMR spectrum (Appendix A) exhibits the vanishing of T^2^ peaks and the presence of a new one at 12 ppm, while the T^3^ signals remained intact between −60 and −70 ppm. This new signal corresponds to the bromotrimethylsilyl group that reacted onto T^2^ substructures. This explains the disappearance of the peaks assigned to T^2^ substructures due to their conversion into T^3^ ones. This means that the structure of the polymer backbone of the networks was not affected by the partial hydrolysis of the phosphonate groups. 

### 3.6. Eu(III) Uptake from Aqueous Medium 

Considering the affinity between phosphonate ligands and trivalent lanthanides, it was of interest to use coordinating copolymers containing organophosphorus ligands as a binding sites for lanthanide separations [102,103,104]. Various studies on that topic have focused on organic–inorganic hybrid Zirconium(IV)−benzene-based-polymers [102], poly(fluorinated acrylate)s and styrene-containing copolymers [103], which showed promise in the removal of different lanthanides including Eu(III) ions. However, to the best of our knowledge, no VDF-containing copolymers used to extract such ions have been reported in the literature. Thus, Eu^3+^ extraction from the polymer network as a function of the lanthanide concentration was studied at 24 °C. The interest in our terpolymers compared to the aforementioned examples lies in the heterogeneous process of extraction since our crosslinked materials cannot be dissolved in any solvents. Various solutions made from different Eu^3+^ concentrations (2, 4, 6, 8, 10 mmol·L^−1^) were prepared in a water/acetic buffer mixture (at pH 5.8) to allow xylenol orange to change its color by complexometric titration [105]. All ion exchange experiments were achieved in batch, with the solid/liquid ratio of 0.25 g of the hydrolyzed crosslinked polymer (**HC****P_20_**)/25 mL of metal ion solution. **HC****P_20_** was chosen for this work because it displays both a strong rigidity and a less swelling affinity in water. Since the crosslinked terpolymer contains 10% of phosphorus atoms only, and assuming that each Eu^3+^ atom requires three ligand groups to lead to a complex [106], the maximum concentration (C_theo_) of the extracted ion could be assessed (3.5 mmol·L^−1^) from Equations (3)–(6), as well as the theoretical uptake (α_theo_%) of each concentration. The results are listed in Table 2.
m = n_1_M_1_ + n_2_M_2_ + n_3_M_3_(3)
n_1_/7 = n_2_/2 = n_3_(4)
n_Eu_ = n_3_/3(5)
C_theo =_ n_Eu_/V_sol_(6)
where m stands for the terpolymer mass (0.25 g), n_1_, n_2_, n_3_ and n_Eu_ are the numbers of mole of VDF, VTEOS, VDMP and theoretical number of moles of Eu extracted respectively, M_1_, M_2_, and M_3_ are the molar masses of VDF (64.04 g·mol^−1^), VTEOS (190.3 g·mol^−1^) and VDMP (136.09 g·mol^−1^), respectively, and V_sol_ is the volume of the prepared solutions (equal to 25 mL).

The Eu^3+^ ion amounts remaining in the liquid phase when reaching equilibrium after 24 h was monitored in the filtrates by complexometric titration with EDTA using xylenol orange as an indicator. Upon reaching the equivalence point, the color of the solution changed from purple to yellow. The uptake degree of the terpolymer at different Eu^3+^ concentrations is represented by Figure 10. 

It is worth noting that pristine PVDF does not show any Eu ion extraction which confirms that metal removal is related to the phosphonic acid function present in the terpolymer [54]. At low metal concentrations (2 mmol·L^−1^), terpolymer **HCP_20_** exhibits a very high ion uptake, evidenced by the absence of any free Eu in solution, which was expected according to the theoretical calculations. As the lanthanide ion concentration increases, a decrease in the uptake capacity is observed. However, this decrease is in agreement with the theoretical results, which indicate that the terpolymer is able to efficiently perform up to 8 mmol·L^−1^. Surprisingly, at 10 mmol·L^−1^ concentration, the obtained Eu extraction was higher than the theoretical one. At this concentration, the immobilization of the lanthanide ions depends on the complexation with the dangling phosphonic acid functions and adsorption in the polymer network [107]. Considering that in the present theoretical calculations, only the effect of complexation is taken into account, at high concentrations, the effect of adsorption becomes more prominent which explains why the terpolymer surpassed the expected extraction values. These data indicate that this crosslinked terpolymer can be employed efficiently for europium extraction from heterogeneous medium, and it may be expected that such results can be extended to other lanthanides and rare earth metals. In comparison to other studies from the literature, Delaunay et al. [108], similar to our system, reported the complete extraction of lanthanides from tap and river waters using crosslinked PMMA. However, the authors only studied the extraction at low lanthanide concentrations and did not show any extraction at high concentrations of metals. Another notable comparison is that to the study reported by Ghannadi-Maragheh’s team [109], who prepared styrene-based ion-imprinted polymers for the extraction of samarium from water. The authors observed 80% of samarium recovery, from a buffer solution (pH = 6) with a of lanthanide concentration of 5 mg/L. This shows that these polymers are comparable in terms of extraction capacity to other examples form literature and can be a suitable alternative.

## 4. Conclusions

Novel functional VDF-based terpolymers for a specific metal extraction application were prepared, characterized, and involved in Eu^3+^ extraction. These terpolymers were synthesized by simple radical terpolymerization of VDF with commercially available VTEOS and VDMP. Appropriate choice of the initiator enabled to reach up to 85% yield. Poly(VDF-*ter*-VDMP-*ter*-VTEOS) terpolymers were produced in various monomer compositions of VDF (ranging from 70 to 80 mol.%), VTEOS (5–20 mol.%) and VDMP (10 mol.%). The resulting waxy terpolymers were crosslinked under acidic conditions to yield solid networks that could not be dissolved in any solvents. Additionally, the crosslinked terpolymers exhibit a slight weaker thermal stability while its degree of crystallinity (*χ*) was slightly higher than that of the uncrosslinked one. The dimethylphosphonate groups were partially hydrolyzed into phosphonic acid functions using bromotrimethyl silane, to yield phosphonic acid containing-poly(VDF-*ter*-VPA-*ter*-VTEOS) terpolymers that exhibited very high europium extraction from water. This study opens new routes to the application of materials for emerging applications in water decontamination.

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
