# Peer review of "Solid–Liquid Europium Ion Extraction via Phosphonic Acid-Functionalized Polyvinylidene Fluoride Siloxanes"

_polymers, 2020, doi:10.3390/polym12091955_

Round 1

Reviewer 1 Report

The manuscript presents the synthesis and charaterization of VDF terpolymers with vinyl silane and phosphonate monomers followed by hydrolysis/crosslinking and Eu ion extraction studies.

The manuscript is interesting, but uneven as far as charaterization goes.  Nonetheless, overall the work is good and recommended for publication with revisions as indicated below.

The authors state line 223: “Indeed, similar to TAPE and TBPPi, DTBPH is a peroxide, but the copolymerization of VDF and VTEOS and the terpolymerization occurred only when that last one was used. Considering the DTBPH structure and compared to monofunctional TBPPi and TAPE, this initiator is difunctional, meaning that the radical density it releases into the reaction medium is double to that of the other initiators for a similar molar percentage77 (Scheme 2). This increase in radical density ensures that this copolymerization could be initiated more efficiently.” 

This is only partially true. In fact, TAPE and TBPPi are actually peroxycarbonates, and the statement is not normalized for temperature.  In reality, all should work at high temperature, as they all contain a tbu-o* fragment, and indeed, DTBPH can generate more radicals/mole.

Table 1.  Where are the molecular weights, PDIs and compositions of these copolymers? Since they are soluble (NMR) all such data should be provided.  Also, what does that yield refer to exactly?  What is the yield vs each of the monomers?  Without compositions no quantitative statements can be made about the crosslinking density and all the effects of these monomers on properties, etc.  The authors state some composition ranges in the abstact but nothing quantitative in the Table, and how was the composition determined.

Figure 1: assignments of peaks at 4.3 ppm, 3.3 ppm (that’s chain transfer to DMC). Where are the initiator chain ends? Mn by NMR?

Figure 2: the NMR is only very superficially assigned.  TT results in 4 peaks, where is CF2-H, where are the monomer connectivity peaks, etc?  Show a comparison with a typical PVDF from free radical polymerization.

The authors ignore the possible acidic hydrolysis of the phosphate esters, including in Figure 5, but realize that it happens in Figure 6 31P-NMR and Fig. 8.

Author Response

The manuscript presents the synthesis and charaterization of VDF terpolymers with vinyl silane and phosphonate monomers followed by hydrolysis/crosslinking and Eu ion extraction studies.

The manuscript is interesting, but uneven as far as charaterization goes.  Nonetheless, overall, the work is good and recommended for publication with revisions as indicated below.

 Response: We thank the reviewer for his/her recommendation and positive feedback.

The authors state line 223: “Indeed, similar to TAPE and TBPPi, DTBPH is a peroxide, but the copolymerization of VDF and VTEOS and the terpolymerization occurred only when that last one was used. Considering the DTBPH structure and compared to monofunctional TBPPi and TAPE, this initiator is difunctional, meaning that the radical density it releases into the reaction medium is double to that of the other initiators for a similar molar percentage77 (Scheme 2). This increase in radical density ensures that this copolymerization could be initiated more efficiently.” 

This is only partially true. In fact, TAPE and TBPPi are actually peroxycarbonates, and the statement is not normalized for temperature.  In reality, all should work at high temperature, as they all contain a tbu-o* fragment, and indeed, DTBPH can generate more radicals/mole.

 Response: We respectably (partly) disagree with the reviewer since these initiators are not peroxycarbonate but peroxides (or perester for TBPPi). The choice of the reaction temperature is usually triggered by the initiator half-life to ensure a consistent supply of moderate amount of radicals all over the reaction time. The nature of the radical generated is also worth noting since carbon centered radical are not efficient to initiate the polymerization of VDF (that is the major comonomer amount in the media) in contrast to Oxygen centered ones, as in the case of TAPE. At 115 °C, according to the supplier’s product data sheet, the half-life of DTBPH is 10 hr, while those of TAPE and TBPPi are less than 0.1 hr. This means that, at such high temperature, TAPE and TBPPi will completely dissociate very fast releasing a high amount of radicals leading to short cooligomers instead of copolymers. In addition, a higher temperature increases the pressure so that the gaseous VDF located in the sky of the autoclave can be pushed toward the liquid phase to react more easily.  In addition, the reactivity of VDF is crucial to enable the “termonomer induced copolymerization” between VDMP and VTEOS which do not copolymerize. This has been clarified in lines 254-258 of the revised manuscript.

Table 1.  Where are the molecular weights, PDIs and compositions of these copolymers? Since they are soluble (NMR) all such data should be provided.  Also, what does that yield refer to exactly?  What is the yield vs each of the monomers?  Without compositions no quantitative statements can be made about the crosslinking density and all the effects of these monomers on properties, etc.  The authors state some composition ranges in the abstact but nothing quantitative in the Table, and how was the composition determined.

 Response: We appreciate these comments and totally understand the reviewer’s concerns. Actually, it is difficult to answer these questions for the several following reasons. First, before crosslinking, the terpolymers are soluble in acetone and in DMF. However, due to the presence of pendant triethoxysilane groups in the polymer, GPC can not be used to determine both the molar masses and the PDIs, since the polymer will get trapped inside the column.

Second, the composition is usually determined via NMR, and since the expected signals assigned from the different comonomers in the terpolymers overlap together (see page 7, line 270), along with the difficulty of the total removal of the solvent due to the highly viscous nature of the air sensitive polymer, the exact composition of the polymer could not be determined. Neither 1H nor 19F NMR spectroscopies enable to supply accurate data on all sequences and differentiate the connectivity (question below): e.g. CH2CF2-CH2CH(R) with R = P(O)(OCH3)2 or Si(OEt)3.

Third, VDF conversion is difficult to assess since its liquid phase may also be trapped in the total product mixture. Furthermore, as the terpolymers were purified by precipitation, neither NMR nor IR analyses of the filtrates were achieved to calculate the VDMP and VTEOS monomer conversions. Indeed, the ethylenic protons in both VDMP and VTEOS overlap; thus 1H NMR is not able to supply any monomer conversions. Hence, the yields supplied in that manuscript were massic yields of the produced terpolymers (i.e., mass of purified and dried terpolymers/sum of all reactants introduced in the autoclave).

Figure 1: assignments of peaks at 4.3 ppm, 3.3 ppm (that’s chain transfer to DMC). Where are the initiator chain ends? Mn by NMR?

 Response: As known, the signal assigned to –CH2CF2-H chain end resulting from the transfer to DMC is well-reported to be between 6.0 ppm and 6.5 ppm while those assigned to PVDF-CH2OC(O)OCH3 resulting from DMC radical that initiates a PVDF chain are ca. 3.3 and 4.3 ppm, respectively (Figure 1, line 275). The 1H NMR signals of initiator chain end should be present in the 1.2 -1.3 ppm range, which overlap with those attributed to CH3 in the pendant triethoxysilane group. Again, knowing the complexity of the assignment, molar masses cannot be calculated.

Figure 2: the NMR is only very superficially assigned.  TT results in 4 peaks, where is CF2-H, where are the monomer connectivity peaks, etc?  Show a comparison with a typical PVDF from free radical polymerization.

Response: As mentioned above, i) –CH2CF2-H chain end gives a broad and complex triplet of triplets centered in the 6.0- 6.5 ppm range and ii) it is difficult to accurately assign both CF2 and CH2 in CH2CF2-CH2CH(R) with R = P(O)(OCH3)2 or Si(OEt)3. By 19F and 1H NMR spectroscopies. A typical PVDF homopolymer shows a quasi similar 19F NMR spectrum as the one reported in our manuscript (for example, references 79, 85, 87-88 have been given, lines 266-268).

The authors ignore the possible acidic hydrolysis of the phosphate esters, including in Figure 5, but realize that it happens in Figure 6 31P-NMR and Fig. 8.

Response: We thank the reviewer for his/her attention and keen eyes. Figure 5 has been modified in the revised manuscript (page 11, line 326) to show the possible hydrolysis of the phosphonate function into phosphonic acid.

Reviewer 2 Report

The manuscript “Solid-Liquid Europium ion extraction Via Phosphonic acid-functionalized Polyvinylidene fluoride siloxanes.” by Dr. Ameduri et al. presents the synthesis of triethoxysilane and dimethyl phosphonate functional vinylidene fluoride containing terpolymers and its applications in Eu ion extraction from water. The manuscript is well-written, interesting and fits with the journal scope. The spectroscopic characterization is well-done and sounds solid. I will indicate publication, but not in its present form. Before approve the publication I would like to address some questions/suggestions to the authors:

1) The authors must present in the introduction why Europium was chose to the detriment of all metal ions.

2) In the Experimental section, room temperature must be specified (25o C?)

3) The 1H NMR data must be presented with two significant digits before comma.

4) The equipment used in the complexometric titration must be presented.

5) Scheme 1 must be revised to fit better quality. Scheme 2 can be presented as SI. It is not necessary to present in the manuscript.

6) Please revise the thermal data presented in Figure 7. The curve seems non continuous. In addition, around 300oC the red curve seems strange.

7) Concerning the proposed application. It is possible to compare the results with actual literature to highlight the potential application of these new materials?

Author Response

The manuscript “Solid-Liquid Europium ion extraction Via Phosphonic acid-functionalized Polyvinylidene fluoride siloxanes.” by Dr. Ameduri et al. presents the synthesis of triethoxysilane and dimethyl phosphonate functional vinylidene fluoride containing terpolymers and its applications in Eu ion extraction from water. The manuscript is well-written, interesting and fits with the journal scope. The spectroscopic characterization is well-done and sounds solid. I will indicate publication, but not in its present form. Before approve the publication I would like to address some questions/suggestions to the authors:

  • The authors must present in the introduction why Europium was chose to the detriment of all metal ions.

Response: We thank the reviewer for his/her recommendation. The introduction has been modified to show why Europium was chosen to represent the possible capabilities of our polymers to remove lanthanides from water. The end of the introduction of the revised manuscript (lines 92-95) states (including five new references) “The resulting materials have been utilized to extract europium ions from aqueous media. The choice of europium as a representative of lanthanides is due to its similar behavior as other lanthanides during extraction73,74 and because it easily forms complexes with xylenol orange,75,76 a commonly used indicator in colorimetric titrations.77

2) In the Experimental section, room temperature must be specified (25oC?)

Response: We thank the reviewer for his/her observation, the manuscript has been modified accordingly (24 °C in lines 120, 177, 188, 364).

3) The 1H NMR data must be presented with two significant digits before comma.

Response: We assume that the reviewer suggests digits after the coma, right? In that case, the 1H NMR data have been modified as requested (in lines 153-161 and 265-277).

4) The equipment used in the complexometric titration must be presented.

Response: The complexometric titration was performed using a classical titration buret. The experimental section has been modified accordingly (lines 201-205). 

5) Scheme 1 must be revised to fit better quality. Scheme 2 can be presented as SI. It is not necessary to present in the manuscript.

Response: We thank the reviewer for his/her comment, the schemes have been modified according the reviewer’s suggestion.

6) Please revise the thermal data presented in Figure 7. The curve seems non continuous. In addition, around 300oC the red curve seems strange.

Response: Probably, the Word to pdf conversion induced some issues. We hope that the revised figure (line 351) is more suitable.

7) Concerning the proposed application. It is possible to compare the results with actual literature to highlight the potential application of these new materials?

Response: The comparison has been inserted in page 14 (lines 440-448), as follows: In comparison to other studies, Delaunay et al.108, similar to our system, reported the complete extraction of lanthanides from tap and river waters using crosslinked PMMA. However, the authors only studied the extraction at low lanthanide concentrations and did not show any extraction at high concentration. Another notable comparison deals with Ghannadi-Maragheh’s team109‘s study that reports the preparation of ion imprinted polystyrene for the extraction of samarium from water. The authors observed 80% of samarium recovery, from a buffer solution (pH =6) with a lanthanide concentration of 5 mg/L. This shows that these terpolymers are comparable in terms of extraction capacity to other examples form the literature and can be a suitable alternative.